



# Wind tunnel testing of a swept tip shape and comparison with multi-fidelity aerodynamic simulations

Thanasis Barlas, Georg Raimund Pirrung, Néstor Ramos-García, Sergio González Horcas, Robert Flemming Mikkelsen, Anders Smærup Olsen, and Mac Gaunaa

DTU Wind Energy, Frederiksborgvej 399, 4000 Roskilde, Denmark

**Correspondence:** Thanasis Barlas (tkba@dtu.dk)

**Abstract.**

One promising design solution for increasing the efficiency of modern horizontal axis wind turbines is the installation of curved tip extensions. However, introducing such complex geometries may move traditional aerodynamic models based on Blade Element Momentum (BEM) theory out of their range of applicability. This motivated the present work, where a

swept tip shape is investigated by means of both experimental and numerical tests. The latter group accounted for a wide variety of aerodynamic models, allowing to highlight the capabilities and limitations of each of them in a relative manner. The considered swept tip shape is the result of a design optimization, focusing on locally maximizing power performance within load constraints. For the experimental tests, the tip model is instrumented with spanwise bands of pressure sensors and is tested in the Poul la Cour wind tunnel at the Technical University of Denmark (DTU). The methods used for the numerical

tests consisted of a blade element model, a near-wake model, lifting-line free-wake models, and a fully resolved Navier-Stokes solver. The comparison of the numerical and the experimental tests results is performed for a given range of angles of attack and wind speeds, which is representative of the expected conditions in operation. Results show that the blade element model cannot predict the measured normal force coefficients, but the other methods are generally in good agreement with the measurements in attached flow. Flow visualization and pressure distribution compare well with Computational Fluid Dynamics

(CFD) simulations. The agreement in the clean case is better than in the tripped case, indicating an aggressive tripping of the flow in the measurements. Some uncertainties regarding the effect of the boundary layer at the inboard tunnel wall and the post stall behavior remain.

## 1   Introduction

The trend of reducing the Levelized Cost of Energy (LCOE) of horizontal axis wind turbines through increasing rotor size has long been established. To achieve this, the challenges of scale must be overcome through innovative turbine design and control



strategies (Veers et al., 2019). One promising blade design concept is advanced aeroelastically optimized blade tip extensions, which could drive rotor upscaling in a modular and cost effective way.

Existing bibliography relevant to wind turbine applications typically focuses on winglets and aerodynamic tip shapes, with limited testing in controlled conditions (Johansen et al., 2006; Gaunaa et al., 2007; Gertz et al., 2012; Hansen et al., 2018). Moreover, there is no relevant research work focusing on details of tip shape aerodynamics relevant to the application of tip extensions for blade upscaling.

In the present work, the aerodynamics of a curved tip shape is investigated via wind tunnel experiments and numerical modeling. The considered swept tip shape is the result of design optimization, focusing on locally maximizing power performance within load constraints compared to an optimal straight tip, for testing in an outdoor rotating test rig (RTR). The tip model is instrumented with spanwise bands of pressure sensors and is tested in the Poul la Cour wind tunnel at the Technical University of Denmark (DTU), for a range of angle of attack and wind speed. Aerodynamic models of different fidelities are utilized to simulate the wind tunnel cases and are compared with the measurement data, namely a blade element model, a near-wake model, lifting-line free-wake models, and a fully resolved Navier-Stokes solver.

## 2 Tip model design

The tip shape presented in this work is the result of an aeroelastic optimization for maximizing power performance within load constraints for a tip mounted on DTU's rotating test rig (RTR) (Madsen et al., 2015; Ai et al., 2019). The optimization method used is the same as the one described in (Barlas et al., 2020), used for the tip design of a full scale wind turbine. The method of optimizing the tip for the RTR is essentially the same, while the baseline geometry and load envelope is defined by a reference straight tip, designed for an optimal BEM performance. The reference tip was designed using the FFA-W3-211 airfoil with fully turbulent wind tunnel polars (Bertagnolio et al., 2001) at a Reynolds number of 1.78e6 (Fig. 1). A predefined length of 3m was used as a design constraint for an outdoor rotating test rig, which is not part of this article and where the tip is mounted on a 8m cylindrical boom. The chord and twist distributions of the straight tip were determined from BEM performance for optimal Cp in operation at 30rpm and 6m/s inflow wind speed. The resulting aeroelastically optimized tip utilizing sweep, achieved a 19.58% increase in power with the same ultimate flapwise bending moment at the boom root and tip connection as the baseline. The design was evaluated with the near wake model in the aeroelastic code HAWC2 (Larsen and Hansen, 2007) for an extreme turbulence case (class III-C) at a wind speed of 6m/s. Compared to the reference straight tip, the design features a highly swept (in-plane offset) centerline (Fig. 2), a slender chord distribution, and a negative twist distribution towards feather (Fig. 3).

The geometry of the optimal tip is scaled with a factor of 0.5 compared to the RTR tip dimensions in order to be accommodated in the Poul la Cour wind tunnel (PLCT) at DTU (Fig. 4). The wind tunnel speed is tuned accordingly in order to achieve the same range of Reynolds numbers compared to operation on the RTR (0.8e6-1.5e6).



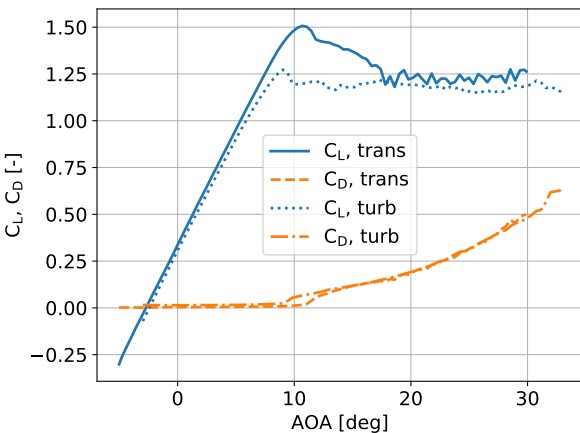

**Figure 1.** Cl and Cd versus angle of attack for the FFA-W3-211 airfoil in free transition and fully turbulent conditions (KTH wind tunnel data (Bertagnolio et al., 2001)).

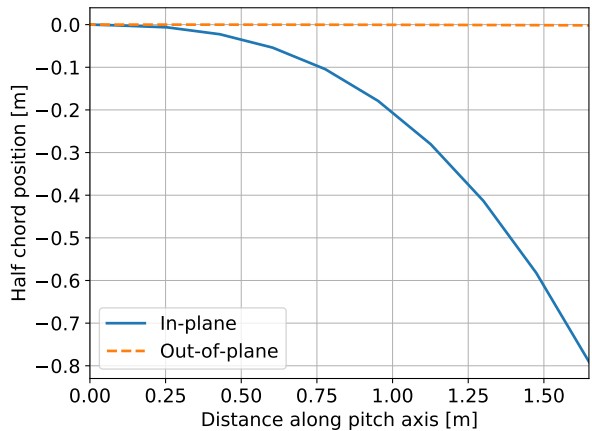

**Figure 2.** Centerline of the tip design.

## 3 Wind tunnel test setup

The PLCT is a closed return tunnel with a closed test section. When testing the tip, the test section uses an aerodynamic setup with hard walls. The rectangular test section has the dimensions of height, H=2.0m, width, W=3.0m and length, L=9.0m. The effective contraction ratio of 9 and the system of screens and Honeycomb results in a low turbulence level of Tu<0.1% for a frequency range of 10–5000 Hz and a flow velocity of 50 m/s. The turntables have a diameter of 1.355m, with a 0.5m x 1.25m hatch with rounded corners. The center of the turntables are located 4 m downstream of the contraction. The tip is mounted in the upper turntable (Fig. 5).





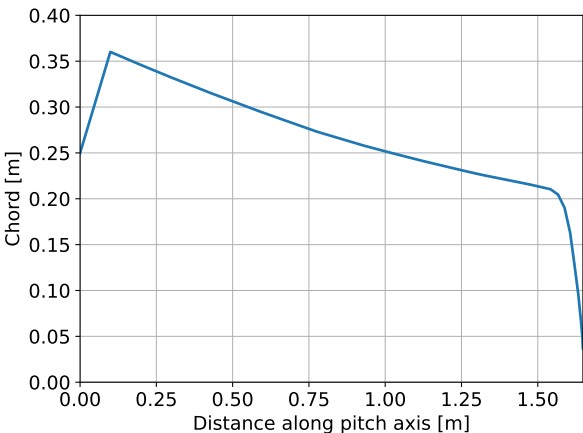 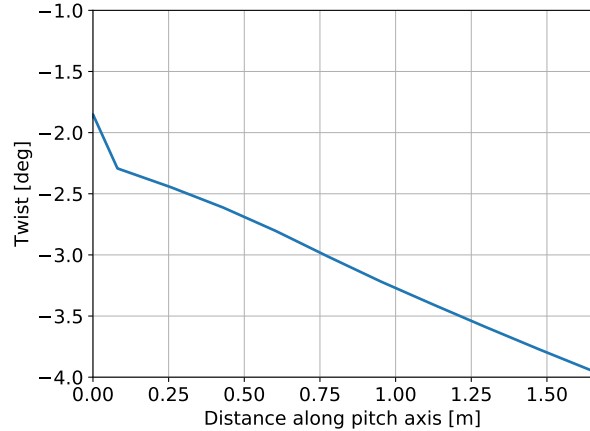

**Figure 3.** Planform of the tip design.

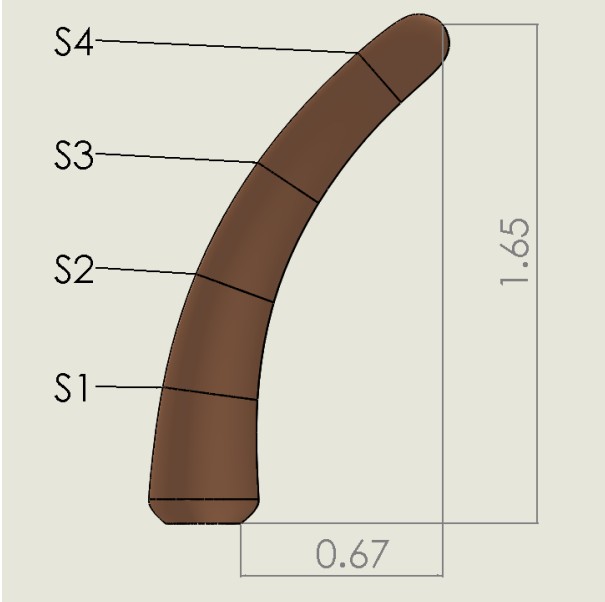

**Figure 4.** 3D geometry of the tip (in meters), indicating the 4 sections where pressure taps are located.

The pressures measured from surface pressure taps in the model are numerically integrated to determine the normal and
tangential force components. The data acquisition (DAQ) system is based upon the CompactRIO system from National Instruments and a DTU in-house made LabView program.

Each of the four sections is equipped with 32 pressure taps. The same normalised chordwise positions are used on all four sections. In the leading edge region (the first 10% of the chord) 14 taps are distributed evenly along the arc length. On the remaining 90% of the chord, nine taps are distributed on each side. Again, they are equally distributed along the arc length.



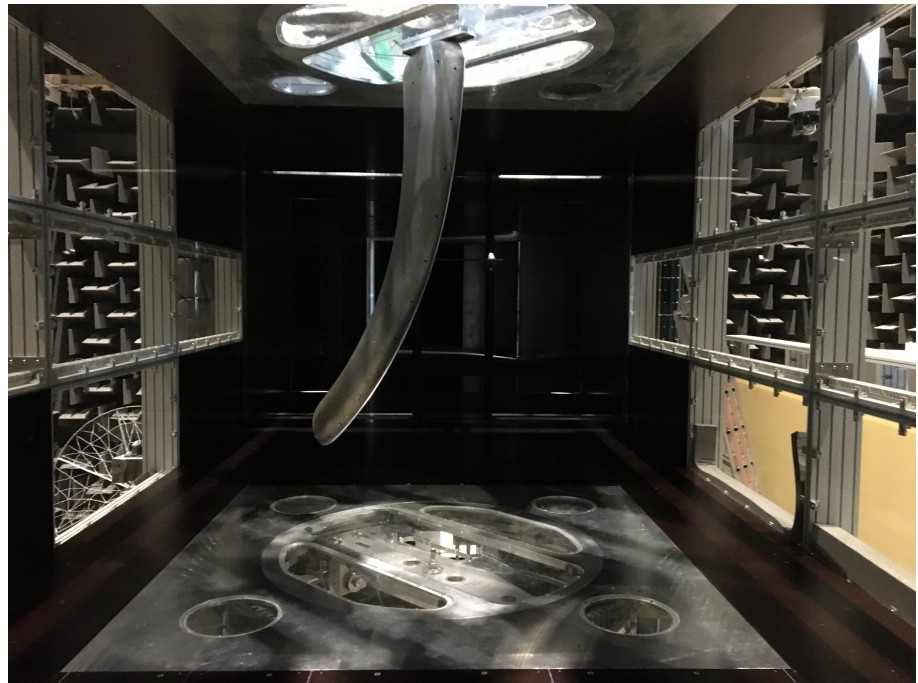

**Figure 5.** The tip model mounted in the test section of the Poul la Cour wind tunnel.

The last tap is located at approximately 90% chord on each side. In the post-processing a point at 100% of the chord is added
where the pressure is assumed to be the average of the pressures at the last tap on each side.

The tip is tested in a range of wind speeds, angles of attack and surface conditions, as shown in Table 1.

| Wind speed [m/s] | Surface condition | AOA range [deg] |
|---|---|---|
| 20 | clean | -180:+180 |
| 40 | clean | -180:+180 |
| 60 | clean | -20:+20 |
| 20 | tripped | -180:+180 |
| 40 | tripped | -180:+180 |
| 60 | tripped | -20:+20 |

**Table 1.** Test configurations. Tripped; zz-tape (0.205mm height, 0.6mm wide, 70 deg) at 5%c on suction side and 10%c on pressure side.

## 4   Numerical simulations

The different aerodynamic models used for the numerical simulations, together with the corresponding setups, are described in
this section. Based on the labels used in the present document, those could be ordered in terms of fidelity as: HAWC2 (blade



element model), HAWC2 near wake, MIRAS (free wake lifting line) and EllipSys3D (CFD). In addition to these models, a different lifting line code, LLTunnel, was utilized as part of this work for evaluating the effect of the wind tunnel, which was not fully included in any of the previous models. In terms of fidelity, LLTunnel could be thought of as lying between HAWC2 near wake and MIRAS, because it is not a free wake method. However, it does model the full interference effect of the tunnel

on the aerodynamic response. Both EllipSys3D and MIRAS correspond to independent fluid dynamics solvers. Those two codes were run in the present study through the external coupling framework referred to as *DTU coupling* (Horcas et al., 2020; Ramos et al., 2020). In this way, the results were integrated in the aeroelastic solution of HAWC2. It should be remarked that the stiff nature of the studied tip made this integration unnecessary from the results point of view. Nevertheless, the use of the DTU coupling framework ensured the consistency of the studied tip geometries, as well as the direct comparability of the

outputs presented in this work. In particular the integrated forces for several cross sections, which were perpendicular to the mid chord line.

### 4.1  Simulated Geometries

The tunnel wall at the root section is 43 centimeters away from the innermost instrumented section on the model. In the tunnel, this wall will 1) prevent the formation of a strong root vortex and 2) create a boundary layer at the root wall which will cause

the velocity to decrease towards zero in its vicinity. To model the first effect of the wall on the trailed vorticity behind the tip, a mirrored tip is simulated in all codes except for LLTunnel. This is achieved by mirroring the tip geometry at the root section in HAWC2 near wake and MIRAS. A flat plane with a symmetry condition is added in EllipSys3D. The blade element method in HAWC2 will not see any effect of mirroring due to the missing cross sectional aerodynamic coupling. In contrast to the other codes, LLTunnel models the effect of all four tunnel walls using the method of mirror images.

### 4.2  HAWC2 and HAWC2 near wake

BEM, which can otherwise be used to compute the induced velocity at a rotor disc, is not applicable in the present study. The single tip configuration resembles more closely a blade in standstill than a rotor in operation. In this case only the blade element part of blade element momentum theory is applicable. The wind speed is projected into the airfoil cross sections, relative velocity and angle of attack are computed and lift, drag and moment coefficients are interpolated from airfoil polar

tables. A tip loss model typically used in BEM is not relevant, because no rotor induction is present, and all the sections are radially independent. Results from this basic blade element approach are labeled 'HAWC2' in the following.

The near wake model, a simplified lifting line model (Pirrung et al., 2016, 2017a), was previously extended to standstill conditions to provide induction modeling where BEM theory is not applicable (Pirrung et al., 2017b). This model computes the cross sectional aerodynamic coupling through the trailed vortex, which will for a single tip mainly result in strong vortices

trailed from the root and tip sections. The near wake model was recently extended to model swept blades in operation (Li et al., 2018), but this extension is not yet available for stand still cases. So the geometry of the wake in the 'HAWC2 near wake' (or simply HAWC2 NW) computations is that of a straight wake behind a straight tip. The relative velocities and AOA at each section are though computed by projecting the wind speed into the airfoil cross sections of the swept tip as in the 'HAWC2'



case described above. The tip was discretized into aerodynamic sections and vortex trailing points at the root, tip and in between
sections according to a cosine distribution.

### 4.3 MIRAS

Simulations with the multi-fidelity vortex solver MIRAS (Ramos et al., 2016, 2017, 2019) have been carried out, using a
built-in lifting line (LL) aerodynamic module in combination with a free-wake filament based model.

In what follows, a description of the LL free-wake model employed in all the HAWC2-MIRAS simulations is detailed. In
the model, the blades are represented by discrete vortex rings along the span. These elements account for the bound vortex
strength and release vorticity into the flow. The bound vortex is discretized with 80 equally spaced straight segments in the
mirrored "c-shaped" configuration. The leading segments of the bound vortex rings are placed along the blade quarter chord
line, with the collocation point located at the three-quarter chord.

The strength of these vortex filaments is calculated via the Kutta-Joukowski theorem, $\underline{\Gamma}$,

$$\overline{\Gamma} = \frac{\overline{L}}{\rho \overline{V}_{cp}} \tag{1}$$

where $L$ is the lift force of each aerodynamic section, obtained by interpolation in a set of tabulated airfoil data ($C_l$,$C_d$,$C_m$) as
function of the computed angle of attack. $\rho$ is the air density at a given temperature and $V_{cp}$ is calculated as follows,

$$\overline{V}_{cp} = \overline{V}_0 + \overline{u}_w + \Delta \overline{u}^b \tag{2}$$

where $\overline{V}_0$ is the free-stream velocity, $\overline{u}_w$ is the wake induced velocity, and $\Delta \overline{u}^b$ accounts for the curved bound vortex
influence as detailed in Li et al. (2020).

The motion of the rest of the filaments is described by Lagrangian fluid markers placed at the filament end points. The
filaments are therefore convected downstream with a velocity, which includes the contribution from the free-stream, the bound
vorticity and the wake induction. The induced velocities are calculated directly by evaluating the Biot-Savart law. To desingu-
larize the Biot-Savart law, the Scully et al. (1972) vortex core profile is applied to all the released vortex filaments. In this way,
an approximation to viscous diffusion, vortex core growth and vortex straining can be included into the inviscid wake model.

Pitch angles from -5 to 20 degrees with a pitch step of 1 degree have been simulated. A time step of 0.001 s is used, with 300
time steps between pitch increments. The total number of filament rows used to represent the wake is fixed at 300, as shown in
Figure 6. A total of 7800 time steps have been computed per simulation. Note that the model is considered rigid in this study.

### 4.4 LLTunnel

The key elements of the LLTunnel lifting line model are essentially identical to those in MIRAS. However, three main points
set LLTunnel and MIARS apart. 1: LLTunnel solves directly for the steady state solution, whereas the MIRAS solution evolves
an unsteady solution. 2: LLTunnel is not a free wake model. The trailed vorticity is assumed to convect downstream directly in
the wind/tunnel direction, whereas MIRAS solves for the time true evolution of the force free wake. 3: MIRAS does not have





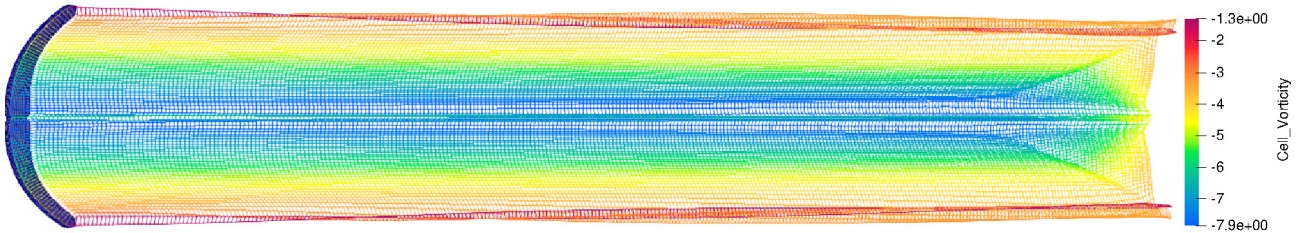

**Figure 6.** MIRAS simulation of the c-shape configuration with the free filament wake model.

the possibility to enforce walls in the domain in its present version, so the effect of the wall on which the wing is mounted is
effectively modeled by modeling also the mirror image of the wing on the other side of the wall. This way there is no flow
through the mirror plane, effectively making it a slip wall. This is treated differently in LLTunnel, where any walls in the
vicinity of the blade simulated in LLTunnel are simulated using mirror images of the blade and wake vorticity when setting up
the influence coefficients of the method. This is also equivalent to enforcing symmetry planes, but it is set up such that more
than one wall can be modelled. The added complexity here is that in this case also the mirror images are mirrored. Simulating
a wing between two walls therefore result in an infinite row of mirrored blades; the vortex equivalent to the visual impact of
standing between two parallel mirrors. In case of two sets of parallel walls, as is the case in the wind tunnel, the result is a full
matrix of mirrored vortex elements. Figure 7 show schematically the mirroring method used in LLTunnel.

In the code only the 20 nearest mirror images in each direction was included. The total number of mirrored vortex systems is
then (20+1+20)X(20+1+20)-1=1680. Using 20 mirror elements to each side was determined as a good number as the difference
when resolving in stead the 30 nearest elements had a negligible influence on the results. The blade, and thereby also all mirror
elements, is discretized used 80 equidistant elements along the blade span for all LLTunnel claculations shown in this work, as
grid studies showed negligible differences in the results for finer resolutions. The effect of point 1 (steady solution) and point 2
(prescribed, non-free wake) is that the method is significantly faster than MIRAS, but that the detailed effects linked to a free
wake is not captured. The effect of this will be shown later when comparing the results of MIRAS and LLTunnel. In the context
of the present paper, LLTunnel will be used only to assess the difference in modeling a blade on a symmetry wall, which is
what is being modelled by all other simulation tools, and modelling a blade in the tunnel, which is what is being measured in
the experiments.

### 4.5 EllipSys3D

Higher fidelity simulations were performed with the three-dimensional computational fluid dynamics code EllipSys3D (Michelsen
, 1992, 1994; Sørensen , 1995). EllipSys3D is a finite volume solver for structured grids, and it implements a wide variety of
turbulent models. In the present study, the incompressible Reynolds-Averaged Navier-Stokes (RANS) equations were solved,
using the k-$\omega$ SST turbulence model (Menter , 1994). Two distinct sets of simulations were performed. One assumed fully
turbulent flow, while the other accounted for a correlation-based transition model (Sørensen , 2009). These two sets of compu-
tations are labelled in the present document as `turb` and `trans`, respectively.



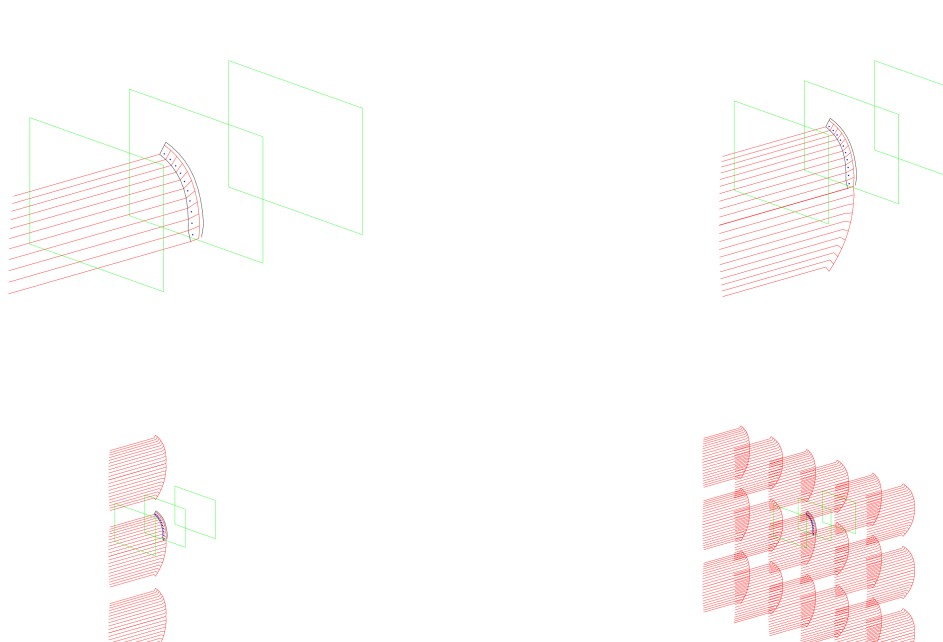

**Figure 7.** Illustration of the use of mirror images model walls. Upper left: Vortex system of physical wing only. The tunnel is outlined in green. Upper right: Vortex system of physical wing and mounting wall vortex systems. This corresponds to what is modelled in all other codes. Lower left: Vortex system of the nearest mirror images from two parallel walls. Lower right: Vortex system from the nearest mirror images due to all four tunnel walls.

A common grid was used for all the EllipSys3D simulations. It was generated in two consecutive steps. First, a structured mesh of the tip surface was generated with the openly available Parametric Geometry Library (PGL) tool (Zahle , 2019). A total of 96 cells were used in the spanwise direction, and the chordwise direction was discretized with 256 cells (with 8 of them lying on the trailing edge). To facilitate the whole grid generation process, the near-root contraction geometry was simplified by assuming a constant chord. Secondly, the surface mesh was radially extruded with the hyperbolic mesh generator Hypgrid

(Sørensen , 1998) to create a semi-spherical volume grid. A total of 128 cells were used in this process, and the resulting outer domain was located approximately 50 m away from the tip. A boundary layer clustering was taken into account, with an imposed first cell height of $1 \times 10^{-5}$ m, in order to target $y^+$ values lower than the unity. The resulting volume mesh accounted for a total of 3.7 million cells. An inlet/outlet strategy was followed for the boundary conditions of the outer limit of the domain. The root plane was modeled as a symmetry boundary condition, and the tip itself as a no-slip boundary condition. A sketch of

the ensemble of the boundary conditions is depicted in Figure 8, together with a visualization of the mesh. Preliminary studies were performed in order to assess the sensitivity of the grid resolution. It was concluded that the considered discretization is suitable for the type of analysis performed in the framework of the present work.





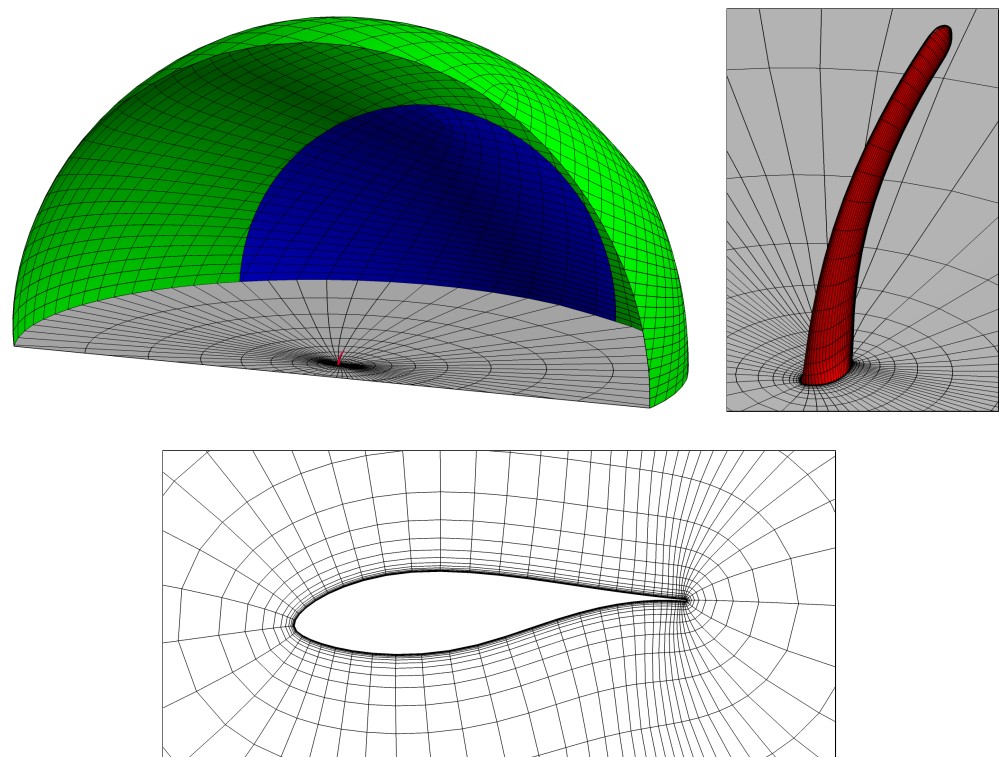

**Figure 8.** Visualization of the EllipSys3D mesh. For clarity, only one out of every four grid lines is shown, and half of the semi-spherical domain is not depicted. Upper row: overview and detail of the boundary conditions distribution (green for inlet, blue for outlet, grey for symmetry and red for wall). Lower row: cross-sectional mesh around the tip shape, taken at one third of is total projected length (starting from the root).

## 5   Comparison of test and simulation results

In this section, the main results of the present work are presented. The first subsection lays the foundation for the rest of
the investigations by quantifying the difference in aerodynamic forces between the blade mounted on a symmetry wall and a blade mounted between four tunnel walls, like the wind tunnel tests. All following sections contain a comparison of test and simulation results in a progressive manner, going from the most qualitative observations to a quantified comparison. In this way, Section 5.2 discusses first the flow patterns around the tip, comparing the experimental tests with EllipSys3D. These observations are complemented by looking at the pressure distributions for both the numerical model and the data acquired
during the experiments (Section 5.3). Finally, Section 5.4 shows a comparison of the sectional loads predicted by each of the numerical methods involved in the present study, where the results obtained from the test campaign are also included.





## 5.1 Assessment of tunnel effects

Before the results from the simulation methods can be compared to wind tunnel measurements, we need to quantify the difference in aerodynamic loading between the wing mounted on a wall, like it is modeled in the majority of the computational
methods employed in the present study, and the wing mounted in the wind tunnel, which is what is being measured in the experiments. This section uses LLTunnel to assess this difference. Figure 9 shows the clean airfoil data simulation results from the lifting-line codes MIRAS and LLTunnel compared for the four blade sections corresponding to the measurement sections in the experiments.

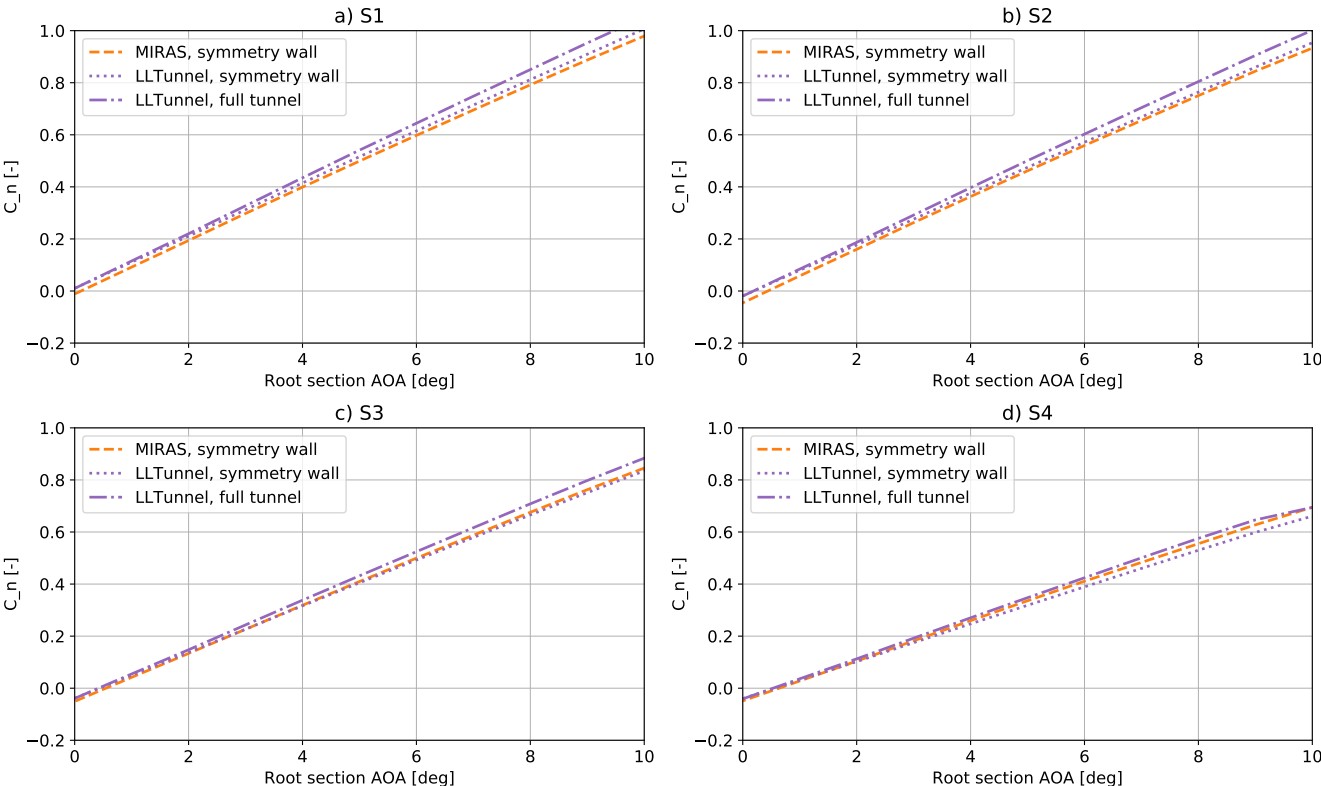

**Figure 9.** The graphs show MIRAS and LLTunnel $C_n$ values as function of root section AOA for blade sections corresponding to the measurement locations on the wing. Upper left: section 1. Upper right: section 2. Lower left: section 3. Lower right: section 4. MIRAS models the wing on a symmetry wall while LLTunnel models both that configuration and the full tunnel configuration.

The results in the figure show that there is a good agreement between LLTunnel and MIRAS results for the single wall
version of the LLTunnel. The relatively small differences between the results can be explained by modeling differences for the two codes. MIRAS includes the free wake effects, which are not included in LLTunnel. On the other hand LLTunnel extends the wake further downstream of the airfoil than MIRAS. The good agreement between the results show that the LLTunnel code is working as intended. The LLTunnel results in the figure also highlights the difference between the wing on a single wall




compared to the wing in the full tunnel setup. The results show that the effect of the tunnel is to increase the normal force
coefficient slightly for all four sections. This is caused by the upwash caused by the additional mirror images in the tunnel case.
The effect of the additional tunnel walls on $C_n$ at all 4 sections is shown in Figure 10. At a root angle of attack of for instance
6 degrees, the increase in $C_n$ is of the order of 0.03 from the single wall mounted wing to the full tunnel mounted wing.

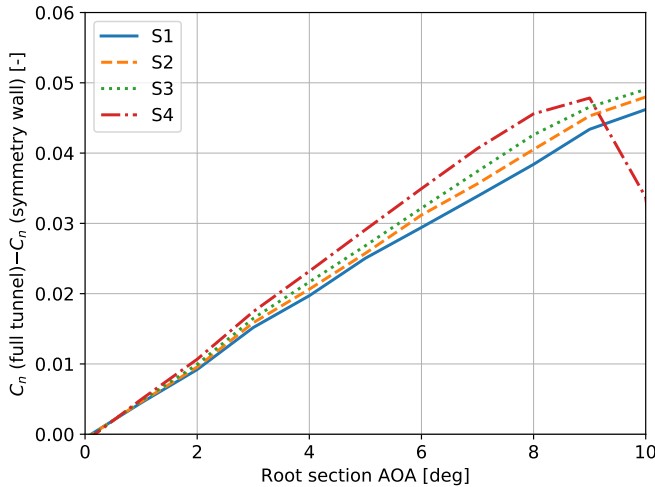

**Figure 10.** The difference in $C_n$ between the full tunnel configuration and the symmetry wall condition. Computed by the LLTunnel code
using clean polars.

Based on this result is is seen that the most of the tunnel effect on the isolated blade is included by simulating the effect
of only the mounting wall, as done in all other simulation tools used in this work. This justifies comparing the results from
the simulation tools to the experimental data. The difference in $C_n$ due to the tunnel is assessed as the difference between the
'tunnel' and 'symmetry' results of the LLTunnel results in Figure 9.

## 5.2    Surface flow

Figure 11 depicts the visualization of the flow around the suction side `SS` of the tip shape, for several angles of attack. Both the
snapshot of the experimental campaign which corresponds to the `clean` configuration, and the `trans` results of EllipSys3D
are presented. For the latter solver, the flow was visualized via surface-restricted streamlines. For the experiments, the record-
ing relied on chord-wise distributed tufts illuminated by UV light. It should be emphasized that this comparison is merely
qualitative, so that the experimental images were not corrected by the camera angle.

At an angle of attack of $0°$, both the experimental results and the numerical model revealed a horizontal flow pattern. When
increasing the AOA to $10°$, some of the trailing edge tufts of the outboard part of the experimental test model showed a slight
vertical component (from root to tip). This feature could be also observed when comparing the streamlines of EllipSys3D at
$0°$and $10°$. Plausible explanations for this effect could be the pressure difference induced by the swept geometry, or the influ-
ence of the tip vortex. Finally, both the experiments and the Navier-Stokes solver predicted stall at $20°$. While the identification

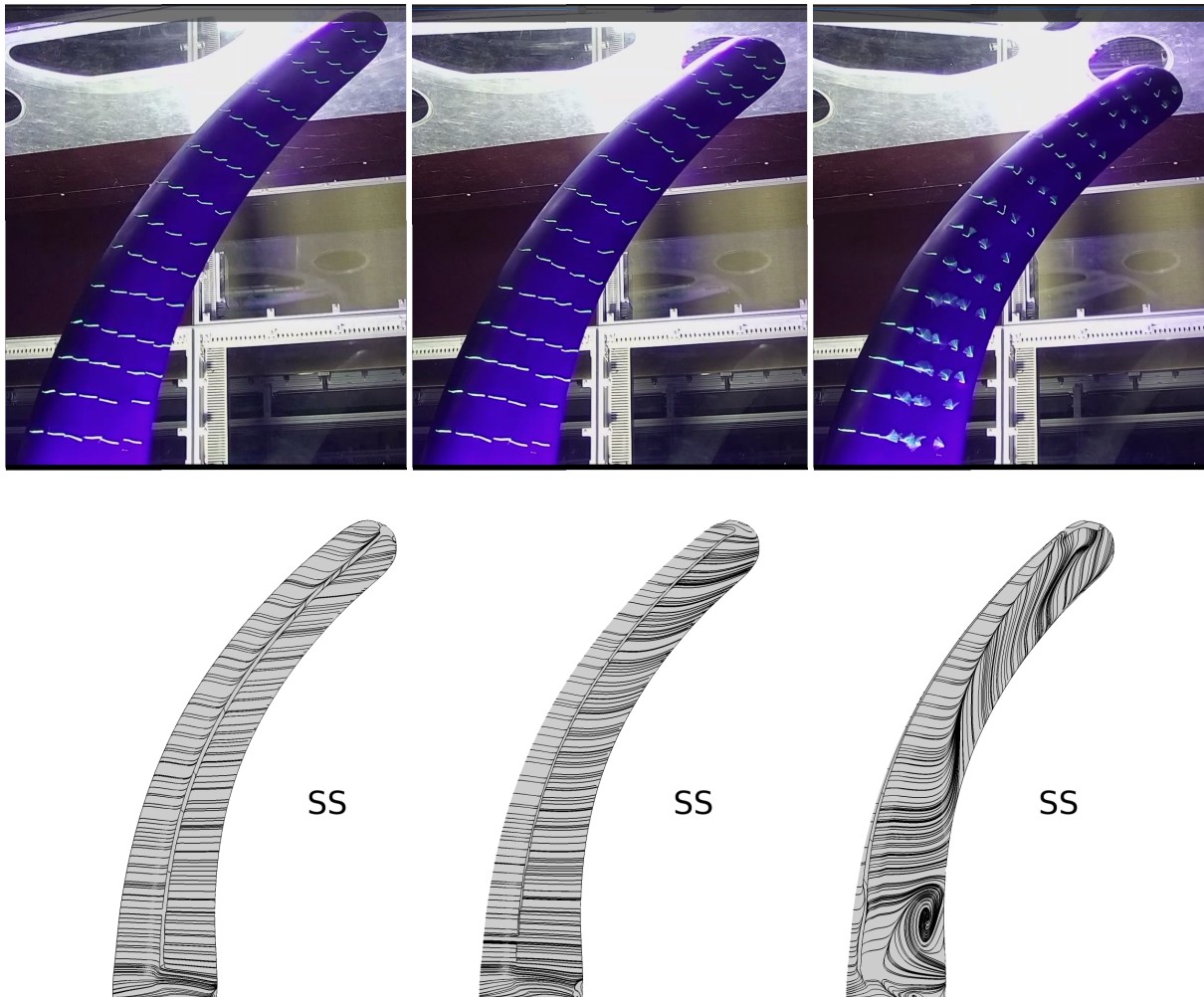

**Figure 11.** Surface flow visualization. Left column: 0°AOA. Middle column: 10°AOA. Right column: 20°AOA. Upper row: experimental results. Lower row: EllipSys3D.

of the separation lines for the former case is not straightforward, those seem to be in agreement with the EllipSys3D prediction. It is then concluded that, from a qualitative point of view, the flow around the tip shape predicted by the Navier-Stokes solver

is in agreement with the observations of the experimental campaign.

### 5.3 Pressure distribution

Figure 12 depicts the comparison of the pressure coefficient distributions for the experimental tests and EllipSys3D. The obtained pressure has been scaled based on the local freestream velocity projection, so that the stagnation corresponded to $Cp = 1$. Overall, a good agreement was obtained, especially between the `clean` experiments and the `trans` simulations.





For the fully turbulent case, the uncertainties related to the installation of the zz-tape on the tip geometry (accurate chord-wise positioning on the 3D geometry) could potentially explain the observed differences. Regarding the different sections, $S1$ is where the discrepancies between the numerical model and the experiments were the highest. That could be explained through the differences in the airfoil geometry at that particular location, since the near-root contraction was replaced by a constant chord evolution in the CFD mesh. While only the 10°angle of attack is included in Figure 12, similar observations could be made for other AOA. It is then concluded that the EllipSys3D predictions are in generally good agreement with the experimental

**Figure 12.** Pressure coefficient $Cp$ distribution, as a function of the normalized chordwise coordinate $x/c$. Each graph corresponds to a different section of the tip shape (see Figure 4). EllipSys3D fully turbulent results `cfd turb`, and with transition model `cfd trans`. For the experimental tests, both clean configuration `exp clean` and tripped `exp tripped` are included.


tests. In Section 5.4, a more quantitative comparison is given by showing the numerical integration of the pressure distributions.





It should be remarked that, due to the relatively poor discretization used for the pressure tabs installation, an integration error should be considered for the experimental results.

## 5.4 Sectional loads

The measured and simulated normal force coefficients are compared in Figure 13 for the clean configuration and in Figure 14 for the tripped configuration. The numerical results are obtained for the wall mounted configuration, a correction for the effect of the remaining tunnel walls (see Sec. 5.1) is not included. To account for this, all simulated results would have to be moved to slightly higher $C_n$, as shown in Fig. 10. All coefficients are normalized by the wind tunnel speed for simplicity, even though the relative velocity at the cross sections differs. The coefficients are shown as function of the angle of attack at the root section.

The local AOA differs due to the twist distribution, see Figure 3.

The results from the pure blade element method denoted 'HAWC2' overpredict the normal loading at all sections. As described in Section 4.2 this is due to the missing cross sectional coupling: any change in slope or post stall levels are only due to the projection of the relative velocities into the airfoil cross sections and the following normalization by the wind tunnel speed. All other codes include aerodynamic cross sectional coupling and thus 3D effects, which lead to reduced slopes at all

sections in both tripped and clean configurations. The HAWC2 NW and MIRAS computations use the same airfoil data. They produce very similar results at the inboard sections, but differ close to the tip due to the larger sweep angles that are ignored in the trailed vorticity computations in the present HAWC2 NW.

The EllipSys3D results in attached flow are in very close agreement with the MIRAS results except for the most outboard region S4, where the slope predicted by EllipSys3D is significantly smaller. This could be explained by the smaller chord

lengths and Reynolds numbers outboard, which lead to worse airfoil performance in EllipSys3D but are not taken into account in the airfoil data input to MIRAS.

In almost all cases EllipSys3D and the measurements both qualitatively predict increased maximum normal force coefficients when comparing to the 2D airfoil data read by HAWC2. An exception is section S2 in the tripped configuration where the maximum measured $c_n$ is below the 2D value. The stall delay seen in the measurements and EllipSys3D results may be

due to the spanwise flow caused by the sweep and proximity to the tip vortex for the outboard sections. Because the EllipSys3D simulations solves the RANS equations, a good representation of the stalled flow region was not expected and thus the behaviour in separated flow will not be discussed further. No 3D correction model for stall delay was used in the codes relying on airfoil data, so also here no accurate prediction of normal force coefficients beyond attached flow is expected.

For the two inboard sections S1 and S2 all models overpredict $c_n$ in the attached flow region in the clean and, much more

pronounced, in the tripped case. In both cases there is some uncertainty due to the boundary layer at the wind tunnel wall close to the blade root, which was not accounted for in the simulations. This boundary layer may cause the loading to drop towards the root, which could cause additional trailed vorticity and reduced $c_n$ slopes. This uncertainty can be addressed in future CFD simulations, where a fully resolved mesh for the whole wind tunnel geometry, including test section, diffuser and nozzle can be simulated. Further, it appears that the flow was tripped quite aggressively in the measurements, see also the comparison of the

pressure distributions in Figure 12, which might cause the airfoil performance in the tripped case to be worse than expected.





As mentioned before, a plausible explanation for the differences between EllipSys3D and the rest of the numerical models was the fact that the latter group used a fixed set of polar data at $Re$=1.78e6. Since the wind tunnel operated at Reynolds numbers between 0.6 and 1.5 millions, that could potentially result in significant discrepancies in the loads prediction. To explore this possibility, the authors performed a side study in order to assess the sensitivity of the lift and drag coefficients with

265 regards to a variation of the Reynolds number within the operational range of the wind tunnel. In particular, polar computations were made with the Navier-Stokes code EllipSys2D (a two-dimensional implementation of the solver used for the present project) and the publicly available software Xfoil (Drela , 1989). Two Reynolds number were considered: 0.8e6 and 1.78e6. For both software, the differences in the predicted load coefficients were considerably smaller than the differences between EllipSys3D and the rest of the numerical codes in the present tip study. As an example, at AOA=5°the Reynolds variation led to

270 relative differences in the order of 2% and -12% for the lift and the drag coefficients respectively. Therefore, it was concluded that the use of a fixed Reynolds number for the polar data did not introduce a significant uncertainty in the results presented in Figures 13 and 14.

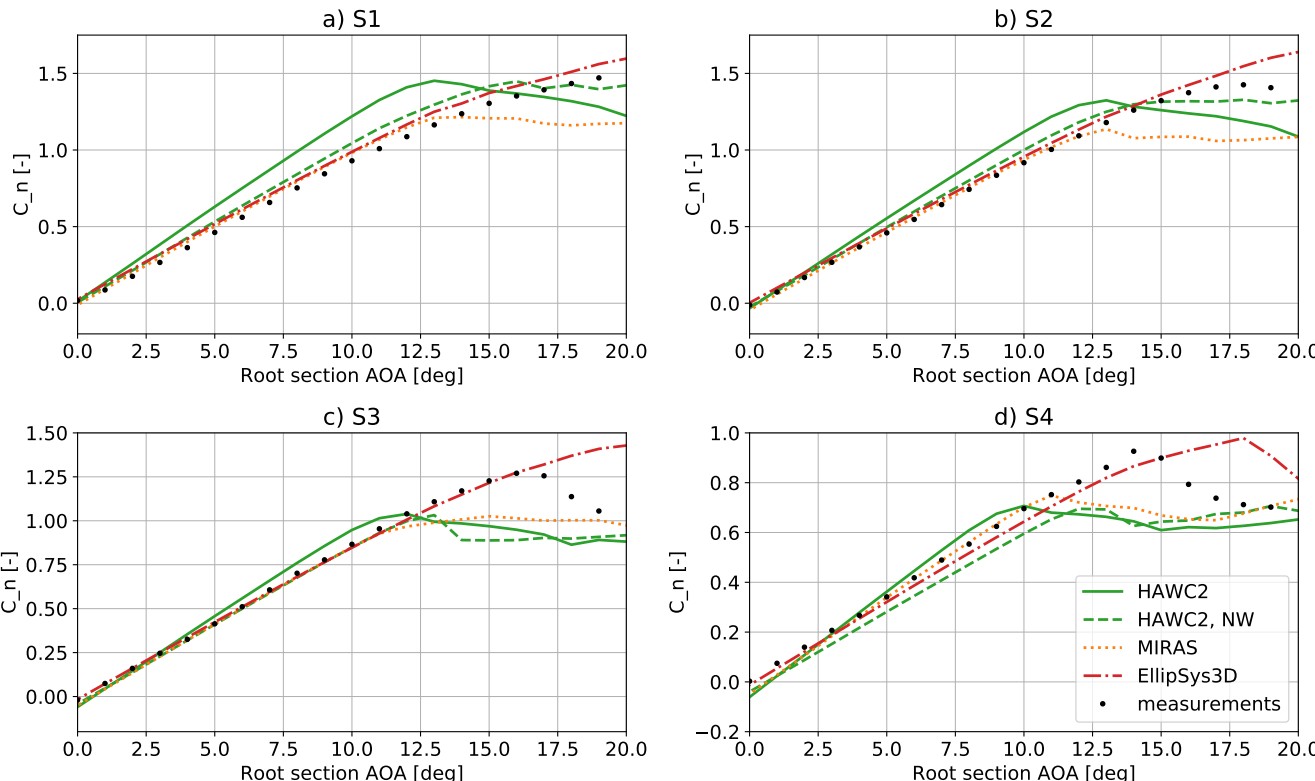

**Figure 13.** Comparison of measured and simulated $c_n$ at the four instrumented sections, clean airfoils.





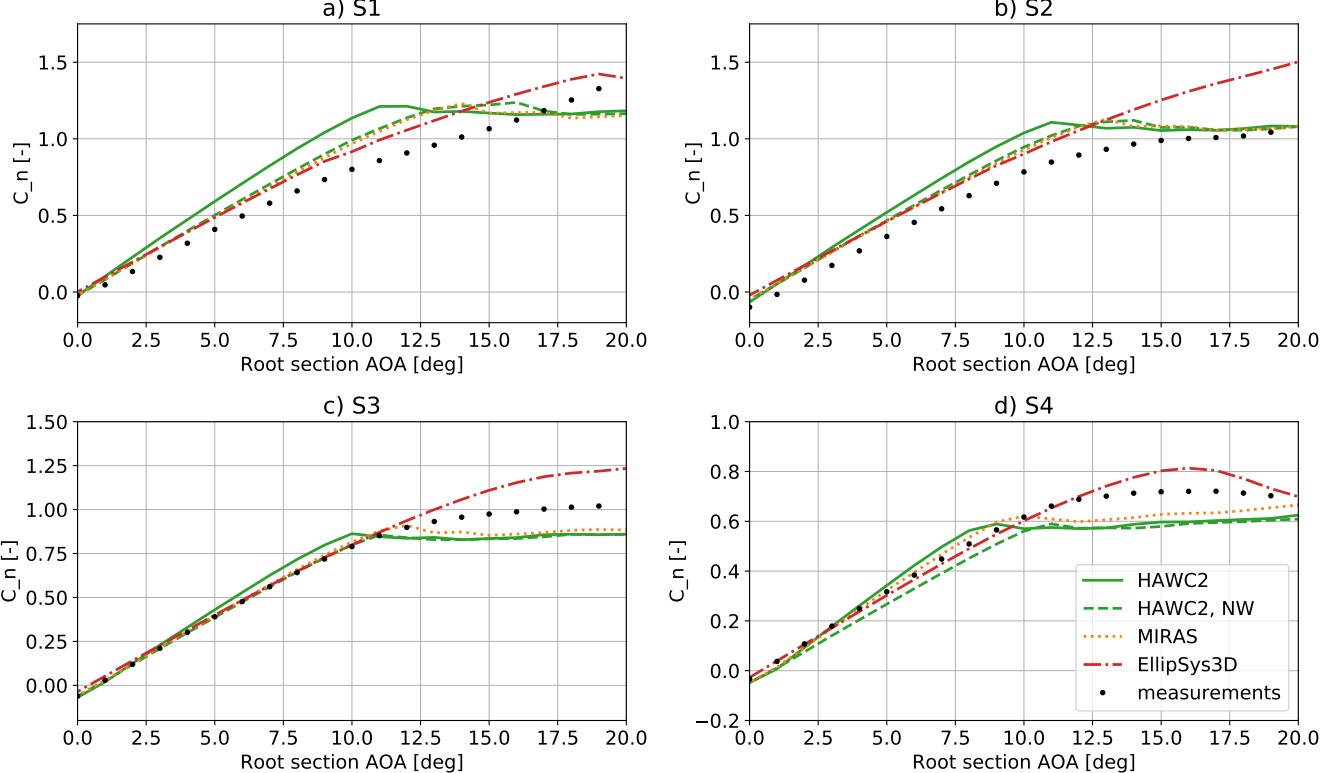

**Figure 14.** Comparison of measured and simulated $c_n$ at the four instrumented sections, tripped airfoils.

## 6    Conclusions

Wind tunnel tests of an optimized swept tip shape are described. A range of fidelity of aerodynamic models is utilized to
simulate the wind tunnel test cases and are compared with the measurement data, namely a blade element model, a near-wake
model, a lifting-line free-wake model, and a fully resolved RANS model. In addition to this, the tunnel effects are assessed with
a different lifting-line code. Results show qualitative agreement of the surface flow in flow visualization and CFD. Comparing
the surface pressure it becomes apparent that there is better agreement for the clean than tripped case, indicating that the zz
tape may have been too aggressive.

When comparing tunnel velocity normalized normal force coefficients as function of geometric root section AOA, important
3D effects cannot be predicted by the blade element model. There is generally good agreement between near wake model,
MIRAS, CFD and experiments in attached flow. However the near wake model predicts the outboard section less accurately
because the curved geometry is not taken into account, and all codes share an uncertainty close to the root due to the neglected
tunnel wall boundary layer. CFD and experiments indicate stall delay, but the quantitative agreement in the post-stall region
is only fair. The clean measurements are generally in better agreement with the simulations than the tripped measurements,





indicating again a too aggressive tripping. Investigations of the tunnel effect show that the $C_n$ values in the tunnel are increased relative to the modelled case of a blade mounted on a symmetry wall. The increase in $C_n$ at a root AOA of 6 degrees is approx 0.03, justifying the direct comparison of the measured data and the simulation results.

Future investigations could focus on clarifying the influence of the wind tunnel wall boundary layer at the root.

*Code and data availability.*   Pre/post-processing scripts and data sets available upon request. The codes HAWC2, MIRAS and EllipSys3D are available with a license.

*Author contributions.*   Thanasis Barlas performed the tip design optimization, wind tunnel model preparation, instrumentation and testing. Georg Pirrung contributed to the tip design optimization and performed model simulations. Néstor Ramos García and Sergio González Horcas performed model simulations. Robert Mikkelsen contributed to the preparation of the experiments and their instrumentation. Anders
Smærup Olsen contributed to the wind tunnel model testing and data post-processing. Mac Gaunaa performed the study on the wind tunnel corrections. All authors contributed to the writing of this manuscript.

*Competing interests.*   No competing interests are present.

*Acknowledgements.*   This research was supported by the project Smart Tip (Innovation Fund Denmark 7046-00023B), in which DTU Wind Energy and Siemens Gamesa Renewable Energy explore optimized tip designs. The following persons have also contributed to the presented
work: Sigurd L. Ildvedsen, Jimmie S. Beckerlee, Helge Aa. Madsen, Flemming Rasmussen, Niels N. Sørensen, Frederik Zahle, Peder B. Enevoldsen and Jesper M. Laursen.



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
