# Peer review of "Wind tunnel testing of a swept tip shape and comparison with multi-fidelity aerodynamic simulations"

_Wind Energy Science, 2021_

## Referee Comment (RC1)

Dear authors,

The article is of high scientific significance and an important contribution to the Wind Energy Science community. The improvement of rotor blades by means of an innovative tip design is investigated, comparing wind tunnel tests with various different simulation methods. The scientific quality is very good. A wide variety of numerical approaches have been applied, taking the important (and tricky) effect of wind tunnel walls into consideration. The measurement methods and results, which are based on the surface pressure distribution could be backed up in some more detail. The study is well-structured and presented in a concise way. However, both introduction/motivation and conclusions should still be improved.

More detailed remarks and suggestions are listed, as follows:

23: „(…) *which could drive rotor upscaling in a modular and cost effective way."* Please clarify what you mean with modular upscaling, and how it is supposed to be cost effective.

26: „*Moreover, there is no relevant research work focusing on details of tip shape aerodynamics relevant to the application of tip extensions for blade upscaling*." It would be helpful to specify the difference between „tip extension" (add-on?) and the „curved tip shape". A graphical representation of a rotor blade with such a curved tip might be helpful as part of the introduction.

30: „*…within load constraints compared to an optimal straight tip, for testing in an outdoor rotating test rig (RTR)."* It is not really clear at this point, whether the "optimal straight tip" and the RTR tests are part of this study or merely considered as next steps, i.e. the outlook of this project. Please specify.

65: „(…) *the average of the pressures at the last tap on each side."* An explanation or reference regarding the validity of this approach would be helpful.

Table 1: Please mention if possible as part of the table 1.) the respective Re numbers 2.) the AoA-steps during the measurements and 3.) a reference or calculation of the ZZ tape-height: is it the same height (0.2mm) along the whole span? Is it 70° or 60° between the serrations of the ZZ tape?

Fig. 7: The graphs are hard to read, especially when printed out.

Fig. 9:

-   How does the simulation of cn evolve for the AoA towards and after stall?
-   The effective local AoA of each section is [AoA(root) minus local twist angle], correct ?

198 -202: "(…) *most of the tunnel effect on the isolated blade is included by simulating the effect of only the mounting wall (…)".* Don't the results (Fig. 9 and 10) contradict this statement, i.e. that including all tunnel walls plays a significant effect on the results?

218: It might be worthwhile mentioning how the resulting local velocity is deducted from the stagnation point (cp=1).

Fig. 12: For readability, please consider presenting clean (trans) and tripped (turb) cases separately. At S4 there seem to be two outliers in the tripped case (red crosses) on the suction side at around 5%c to 10%c. Is it possible that the ZZ tape is interfering with these pressure tabs, e.g. by covering the holes?

226: *"It should be remarked that, due to the relatively poor discretization used for the pressure tabs installation, an integration error should be considered for the experimental results."* The amount of sensors seems rather sufficient. A literature reference would be helpful in terms of the amount of

pressure tabs. Did you perform an estimation of Standard Deviation or uncertainty of the pressure values?

234: *"The coefficients are shown as function of the angle of attack at the root section. The local AOA differs due to the twist distribution, see Figure 3."* It would be helpful to have this explanation already earlier on, e.g. when looking at Fig. 3.

256: Side note: We have had similar problems with airfoil measurements. The effect of the wall boundary layer might be alleviated by attaching one (or two) VGs close to the root.

259: "*Further, it appears that the flow was tripped quite aggressively in the measurements, see also the comparison of the pressure distributions in Figure 12, which might cause the airfoil performance in the tripped case to be worse than expected.*" A simulation of the local boundary layer thickness versus the ZZ tape-height could clarify this issue. Also, the ZZ tape seems to work fine for S3 and S4. Besides, is the specific 3D effect of ZZ tape simulated in any of the codes?

269: "*As an example, at AOA=5 the Reynolds variation led to relative differences in the order of 2% and -12% for the lift and the drag coefficients respectively.*" -12% of drag variation sounds like a relevant Re-number dependency, even though the 2% on lift might be negligible.

Fig. 13 and 14: Is there a reason for not presenting…

-   the LLTunnel results?
-   some of the post stall results in reference to Table 1 (AoA=-180/180°)?

286: The comparison in the tripped case looks OK at S3 and S4.

289: The conclusions end rather abruptly. "*Future investigations (…)*": it would be interesting to close the loop to the introduction and mention next steps e.g. in terms of field tests (RTR) as well as an outlook of whether you consider the research of the curved tip design to be promising compared to conventional or straight tip designs.

---

## Author Comment (AC1)

**Authors' response to review of article wes-2021-48-RC1/RC2**

**RC1**

**RC1_1:**

The article is of high scientific significance and an important contribution to the Wind Energy Science community. The improvement of rotor blades by means of an innovative tip design is investigated, comparing wind tunnel tests with various different simulation methods. The scientific quality is very good. A wide variety of numerical approaches have been applied, taking the important (and tricky) effect of wind tunnel walls into consideration. The measurement methods and results, which are based on the surface pressure distribution could be backed up in some more detail. The study is well-structured and presented in a concise way. However, both introduction/motivation and conclusions should still be improved.

**AC1_1:**

The authors would like to thank the reviewer for their time and greatly appreciate their feedback and suggestions to improve the article.

**RC1_2:**

23: „(…) which could drive rotor upscaling in a modular and cost effective way." Please clarify what you mean with modular upscaling, and how it is supposed to be cost effective.

**AC1_2:**

Given the fact that wind turbine manufacturers nowadays offer modular platform options for facilitating site-specific sales, different tip designs are a potential solution, with reduced investment cost compared to a family of blades. Additional text has been added to highlight this.

**RC1_3:**

26: „Moreover, there is no relevant research work focusing on details of tip shape aerodynamics relevant to the application of tip extensions for blade upscaling." It would be helpful to specify the difference between „tip extension" (add-on?) and the „curved tip shape". A graphical representation of a rotor blade with such a curved tip might be helpful as part of the introduction.

**AC1_3:**

Most of the published work on tips shapes for wind turbine blades focuses on small tip modifications (mainly winglets) which only modify the tip vortex characteristics. This work focuses on aerodynamics of blades with generalized curved shapes. A figure from a previous publication on design of tip extensions for wind turbine blades has been added.

**RC1_4:**

30: „…within load constraints compared to an optimal straight tip, for testing in an outdoor rotating test rig (RTR)." It is not really clear at this point, whether the "optimal straight tip" and the RTR tests are part of this study or merely considered as next steps, i.e. the outlook of this project. Please specify.

**AC1_4:**

The tested tip shape in this work is linked to a full scale aeroelastic tip prototype tested outdoor in DTU's rotating rig. The publication will be submitted soon to WES.

**RC1_5:**

65: „(…) the average of the pressures at the last tap on each side." An explanation or reference regarding the validity of this approach would be helpful.

**AC1_5:**

The issue is that the profile pressure distribution needs to be closed in order to calculate the normal and tangential forces. Hence, an extrapolation method is needed when there is no tap at the trailing edge. The present method is used as a first estimation due to its simplicity and robustness. In any case, the influence from the extrapolation method on the normal force is minor, whereas the effect on the tangential force can be larger, but still within acceptable limits.

A more thorough explanation is added in the paper.

**RC1_6:**

Table 1: Please mention if possible as part of the table 1.) the respective Re numbers 2.) the AoA- steps during the measurements and 3.) a reference or calculation of the ZZ tape-height: is it the same height (0.2mm) along the whole span? Is it 70° or 60° between the serrations of the ZZ tape?

**AC1_6:**

The same zz-tape (0.205mm high, 6mm wide and 70 deg) is used along the entire span.

The table is updated with the requested information.

**RC1_7:**

Fig. 7: The graphs are hard to read, especially when printed out.

**AC1_7:**

The figure has been improved.

**RC1_8:**

Fig. 9:

- How does the simulation of cn evolve for the AoA towards and after stall?

- The effective local AoA of each section is [AoA(root) minus local twist angle], correct ?

**AC1_8:**

- CFD overpredicts Cn as generally expected, while the rest of the codes, relying on 2D data, predict an earlier sharper stall behavior.

- The root geometric AoA is used in the comparison plots as described in the text. The effective local AoA (which is not measured) is a result of the root AoA, twist, transformed velocity components to the local chord and local induction.

**RC1_9:**

198 -202: "(…) most of the tunnel effect on the isolated blade is included by simulating the effect of only the mounting wall (…)". Don't the results (Fig. 9 and 10) contradict this statement, i.e. that including all tunnel walls plays a significant effect on the results?

**AC1_9:**

The Cn difference between the full wall effect and the effect of only the mounting wall (Fig.9 and Fig. 10), is small in terms of absolute numbers. The text has been rephrased to clarify that.

**RC1_10:**

218: It might be worthwhile mentioning how the resulting local velocity is deducted from the stagnation point (cp=1).

**AC1_10:**

This has been clarified in the text by replacing the statement:

"The obtained pressure has been scaled based on the local freestream velocity projection, so that the stagnation corresponded to $Cp = 1$"

with:

"The obtained pressures have been scaled based on the local freestream velocity. Its value was found by forcing $Cp$ to be 1.0 for the lowest pressure of each section."

**RC1_11:**

Fig. 12: For readability, please consider presenting clean (trans) and tripped (turb) cases separately. At S4 there seem to be two outliers in the tripped case (red crosses) on the suction side at around 5%c to 10%c. Is it possible that the ZZ tape is interfering with these pressure tabs, e.g. by covering the holes?

**AC1_11:**

The combined trans/turb data in the figure is still considered fine for presentation purposes The ZZ tape is certainly interfering with the closest pressure taps.

**RC1_12:**

226: "It should be remarked that, due to the relatively poor discretization used for the pressure tabs installation, an integration error should be considered for the experimental results." The amount of sensors seems rather sufficient. A literature reference would be helpful in terms of the amount of pressure tabs. Did you perform an estimation of Standard Deviation or uncertainty of the pressure values?

**AC1_12:**

The number and placement of the pressure taps had been verified in terms of integration error at the design phase of the experiment. The sentence has been removed, since it may create some confusion.

The pressures are measured with Scanivalve MPS4264 scanners with full scale ranges from 6.9kPa to 69 kPa (the highest ranges are used close to the leading edge). The accuracy for all scanners is 0.06% of the full scale range (0.0041 kPa to 0.041 kPa). Our experience is that the actual accuracy is much better, especially for the higher ranges. In a previous study, we looked at the standard deviation of the pressures and found that for attached flow, it is small and we assume this is the case for these measurements as well.

Text has been added to highlight these details.

**RC1_13:**

234: "The coefficients are shown as function of the angle of attack at the root section. The local AOA differs due to the twist distribution, see Figure 3." It would be helpful to have this explanation already earlier on, e.g. when looking at Fig. 3.

**AC1_13:**

This explanation has now been included earlier when referring to Fig. 3.

**RC1_14:**

256: Side note: We have had similar problems with airfoil measurements. The effect of the wall boundary layer might be alleviated by attaching one (or two) VGs close to the root.

**AC1_14:**

This is certainly the common practice for 2D airfoil sections. Different options were considered during the test design with preliminary CFD simulations, and for the particular root shape, it was decided to not add any tripping mechanisms.

**RC1_15:**

259: "Further, it appears that the flow was tripped quite aggressively in the measurements, see also the comparison of the pressure distributions in Figure 12, which might cause the airfoil performance in the tripped case to be worse than expected." A simulation of the local boundary layer thickness versus the ZZ tape-height could clarify this issue. Also, the ZZ tape seems to work fine for S3 and S4. Besides, is the specific 3D effect of ZZ tape simulated in any of the codes?

**AC1_15:**

The authors agree that the implications of the ZZ tape geometry on the flow were not explicitly included in the performed CFD simulations, as they were just included through the assumption of a fully turbulent flow. While the kind of detailed analysis that the reviewer suggests is outside the scope of the present work, we have found this remark to be particularly relevant, and the following comment was added in the text:

"Additionally, it should be reminded that the effects of the ZZ tape were included in the CFD simulations through the assumption of a fully turbulent flow. This could omit some three-dimensional effects related to the particular geometry of the employed tape."

The previous comment is now removed from the text.

**RC1_16:**

269: "As an example, at AOA=5 the Reynolds variation led to relative differences in the order of 2% and -12% for the lift and the drag coefficients respectively." -12% of drag variation sounds like a relevant Re-number dependency, even though the 2% on lift might be negligible.

**AC1_16:**

The % differences (especially in Cd) can indeed be considered significant, but probably still minor considering the absolute coefficient numbers. The sentences has been re-phrased.

**RC1_17:**

Fig. 13 and 14: Is there a reason for not presenting…

- the LLTunnel results?

- some of the post stall results in reference to Table 1 (AoA=-180/180°)?

**AC1_17:**

- The LLTunnel results were utilized for exploring the significance of the tunnel wall effects, and based on the findings it does not seem relevant to include them considering the presentation clarity of the figures.

- The 360deg data were initially considered for standstill vortex shedding cases comparison with CFD, but it was disregarded in this article as being outside the focus of the investigation.

**RC1_18:**

286: The comparison in the tripped case looks OK at S3 and S4.

**AC1_18:**

The statement has been re-phrased.

**RC1_19:**

289: The conclusions end rather abruptly. "Future investigations (…)": it would be interesting to close the loop to the introduction and mention next steps e.g. in terms of field tests (RTR) as well as an outlook of whether you consider the research of the curved tip design to be promising compared to conventional or straight tip designs.

**AC1_19:**

Text has been added to better link the findings to the introduction and future work.

**RC2**

**RC2_1:**

The article is of high scientific significance and an important contribution to the Wind Energy Science community. The improvement of rotor blades by means of an innovative tip design is investigated, comparing wind tunnel tests with various different simulation methods. The scientific quality is very good. A wide variety of numerical approaches have been applied, taking the important (and tricky) effect of wind tunnel walls into consideration. The measurement methods and results, which are based on the surface pressure distribution could be backed up in some more detail. The study is well-structured and presented in a concise way. However, both introduction/motivation and conclusions should still be improved.

**AC2_1:**

The authors would like to thank the reviewer for their time and greatly appreciate their feedback and suggestions to improve the article.

**RC2_2:**

The paper deals with a simulation and wind tunnel campaign of a swept blade tip. Different simulation codes of different complexity and fidelity level are compared against the wind tunnel results. The employed simulation tools are all at the state of the art. The experimental tests were designed in a proper way, and the experimental rig was equipped with a sensor suite adequate for the scope of the investigation. The amount of work done is huge, and the Authors did a great job in summarizing the approach and the results in less than 20 pages, without penalizing the informative content of the paper. The quality of the paper is high but the question on the possible usage of the results (given the fact that the blade tip is fixed and does not rotate).

**AC2_2:**

The authors believe that the possible usage of the results (even in non-rotating conditions) is still considerable. Indeed, there is no prior experience, to the authors' knowledge, assessing different numerical tool fidelities for capturing curved shape effects in the framework of realistic wind turbine tips. Therefore, the authors consider that the presented work is shedding some light around the physics involved and anticipating potential challenges. That being said, the authors agree in the importance of assessing the rotational effects on the capabilities of the methods, and indeed are working in an upcoming article where the full scale version of the tip has been studied in a full rotating test (also in field conditions).

**RC2_3:**

In the introduction, the Authors referred to the literature belonging to the wind energy field and assessed the innovative content of the work according to that. However, I believe that experiments and simulations for wing tips were performed in the past to study the aerodynamics of aircraft wings with different tip shapes. Since the experiments of the present paper consider a fixed (non-rotating) blade, the comparison with the "aircraft world" may be appropriate.

**AC2_3:**

Relevant aircraft related references have been included. The authors still consider that wind turbine specific applications are more relevant, despite the fact that this article focuses on a non-rotating setup.

**RC2_4:**

In the introduction and the conclusions (in general, in the whole paper) a possible extension of the present results to the case of rotating blades is missing. I understand that performing such tests is extremely difficult, but the question is important. Do we expect a similar agreement also for rotating blade tips? This question is even more important for BEM codes, which are typically used for the design and certifications of new machines.

**AC2_4:**

The presented work is considered a stepping stone to the full rotating test (also in field conditions), which will be published in an upcoming article. Text has been modified in the introduction and conclusions to emphasize that.

**RC2_5:**

Often, the swept blades or swept blade tips are responsible for an aeroelastic coupling with blade torsion. In practice, the lift of the swept tip, being behind the shear center of the root, entails a torsional moment which tends to reduce the angles of attack of the blade sections. Since this is an important aspect for load computations (even for mitigation of turbulence loads), if relevant and possible, I encourage the Authors to extract from the experimental data also the total torsional moment at the root of the blade tip and compare it with simulation results.

**AC2_5:**

The authors are aware of this important aspect of swept tips, but the presented tests focus on stiff conditions, in order to isolate the purely aerodynamic contributions. The full aeroelastic behavior of swept tips is analyzed in more detail in an earlier publication (https://doi.org/10.5194/wes-6-491-2021)

**RC2_6:**

Section 2: "The wind Tunnel speed is tuned accordingly in order to achieve the same range of Reynolds numbers compared to operation on the RTR (0.8e6-1.5e6)". Please add if relevant the details (e.g. real and experimented relative airflow velocity, errors between the wind tunnel and the real Reynolds). It could be even interesting to give an indication on Mach number, even though I believe that the flow will be uncompressible with excellent approximation.

**AC2_6:**

Details on the full scale and wind tunnel model Reynolds number calculations have been added.

An indication of the (low) Mach number is also included.

**RC2_7:**

Figure 4: the paper could benefit from the addition in this figure of the points where the pressure taps are located.

**AC2_7:**

This is more clearly shown in Fig. 12.

**RC2_8:**

Pag. 6, line 100 and subsequent ones. It seems that there is a modeling issue with HAWC2 NW and the swept blade. This might jeopardize the goodness of the results. Please comment thoroughly. Is there a reference for LLTunnel?

**AC2_8:**

The authors have previously documented the benefits of the improved NW model for curved geometries (see https://doi.org/10.5194/wes-6-491-2021), but the standstill implementation was not available at the time this work took place.

A reference for LLTunnel has been added.

**RC2_9:**

Pag. 12, line 195: Please, rephrase in order to avoid the repetition of "caused".

**AC2_9:**

The sentenced has been re-phrased.

**RC2_10:**

Pag. 12, line 198: Check: "Based on this result is is seen"

**AC2_10:**

The typo has been corrected.

**RC2_11:**

Fig. 11: It is not clear the meaning of the longitudinal line ranging from root to tip in the visualization of EllipSys3D results. Even looking at the experimental flow visualization (consider for simplicity the 0 deg case), it seems that the flow should be mainly characterized by a chordwise direction, but EllipSys seems to predict a significant spanwise flow component at about half chord location in the 0 deg case, and 40% chord location in the 10 deg case. Maybe, I did some mistakes in interpreting the picture, but, if possible, clarify.

**AC2_11:**

The spanwise line in the computations indicates the boundary layer transition. Both the test and the CFD visualizations indicate some spanwise flow at the outboard sections. Quantifying the differences in the flow just by observing the tufts is though considered quite challenging.

**RC2_12:**

Pag. 16, line 263: I may suggest that the impact of Reynolds number variation is more important at the stall in the CL-alpha curve. Typically, the higher the Reynolds, the higher the stall angle. Maybe, this consideration could be useful to have an improved interpretation of the results.

**AC2_12:**

This is true. Additional text has been added to emphasize that.